# VideoGLUE: Video General Understanding Evaluation of Foundation Models

## Abstract

We evaluate existing foundation models' video understanding capabilities using a carefully designed experiment protocol consisting of three hallmark tasks (action recognition, temporal localization, and spatiotemporal localization), eight datasets well received by the community, and four adaptation methods tailoring a foundation model (FM) for a downstream task. Moreover, we propose a scalar VideoGLUE score (*VGS*) to measure an FM's efficacy and efficiency when adapting to general video understanding tasks. Our main findings are as follows. First, task-specialized models significantly outperform the six FMs studied in this work, in sharp contrast to what FMs have achieved in natural language and image understanding. Second, video-native FMs, whose pretraining data contains the video modality, are generally better than image-native FMs in classifying motion-rich videos, localizing actions in time, and understanding a video of more than one action. Third, the video-native FMs can perform well on video tasks under light adaptations to downstream tasks (e.g., freezing the FM backbones), while image-native FMs win in full end-to-end finetuning. The first two observations reveal the need and tremendous opportunities to conduct research on video-focused FMs, and the last confirms that both tasks and adaptation methods matter when it comes to the evaluation of FMs. We will release our code upon acceptance.

## 1 Introduction

Foundation models (FMs) are a term coined by Bommasani et al. (Bommasani et al., 2021), referring to "any model that is trained on broad data that can be adapted (e.g., finetuned) to a wide range of downstream tasks." Some representative FMs include but are not limited to BERT (Devlin et al., 2018), GPT-3 (Brown et al., 2020), CLIP (Radford et al., 2021), and ALIGN (Jia et al., 2021). This work primarily investigates the video understanding capabilities of six visual and multimodal FMs: CoCa (Yu et al., 2022), CLIP (Radford et al., 2021), FLAVA (Singh et al., 2022), VideoMAE (Tong et al., 2022), VATT (Akbari et al., 2021), and InternVideo (Wang et al., 2022b). These models are selected because they are amendable for the video understanding of our interest and make their checkpoints accessible to us.

It is nontrivial to evaluate FMs. In contrast to "specialist" models developed for a particular task, FMs are considered as "generalists" that learn shareable meta-knowledge across tasks so that one can quickly adapt them to achieve superior performance on various downstream tasks. Hence, *both the tasks and adaptation methods matter when it comes to evaluation.* However, the community has not reached a consensus on these two aspects. FM developers select their own different sets of downstream tasks — interestingly, often covering no video or only appearance-rich video classification tasks (Buch et al., 2022; Lei et al., 2023). Moreover, they rely on distinct adaptation methods, making apples-to-apples comparisons challenging and causing mismatches with the FMs' actual use cases.

To this end, we propose to evaluate FMs' video understanding capabilities using a carefully designed experiment protocol, named VideoGLUE, consisting of three hallmark tasks (action recognition, temporal localization, and spatiotemporal localization), eight datasets well received by the research community, and four model adaptation methods tailoring a foundation model for downstream tasks. The tasks examine an FM from various aspects needed for understanding video. The "all-around" adaptations represent the main use cases of FMs in the literature and, more importantly, allow us to thoroughly probe an FM's potential in video understanding.

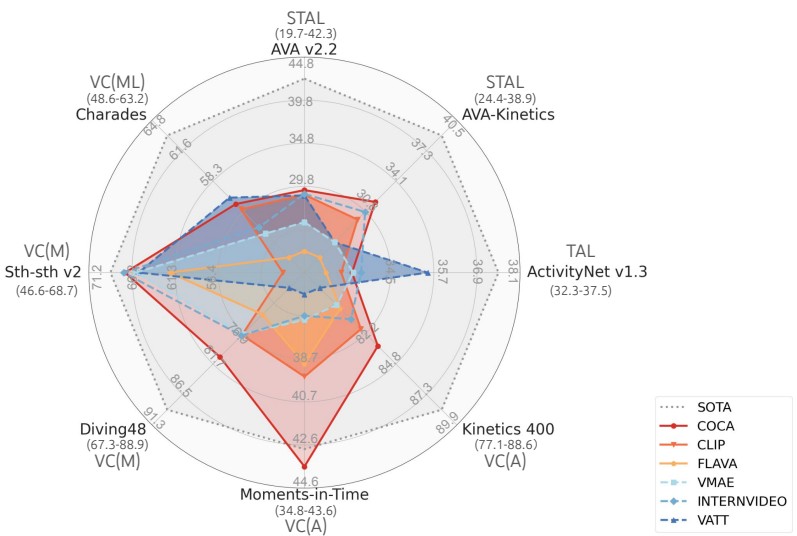

Figure 1: FMs vs. state-of-the-art task-specialized models on video understanding. Unlike natural language and image understanding, video tasks are where FMs generally fall behind "specialists". VC(A), VC(M), and VC(ML) stand for appearance-focused, motion-focused, and multi-labeled video classification, respectively. STAL stands for spatiotemporal action localization, and TAL stands for temporal action localization. For each task, we include the (min-max) range shown in the figure.

Why do we specifically focus on videos? The main motivation is to promote video understanding in the evaluation of FMs. More concretely, we test the following conjectures through this work. First, FMs' high performance on existing evaluation suites does not necessarily indicate their potential in video since these suites either lack video-specific tasks or selectively choose video tasks whose appearance feature is more important than motion — InternVideo (Wang et al., 2022b) is an exception as discussed in the next paragraph. Second, many existing FMs probably cannot heed motion in video, given that they learn primarily from static images (Radford et al., 2021; Singh et al., 2022; Yu et al., 2022) or short video clips containing limited motion (Feichtenhofer et al., 2022; Wang et al., 2022b). Third, popular adaptation methods (e.g., finetuning all weights) cannot supplement FMs with all the cues needed to recognize motion-rich actions and localize entities temporally and/or spatiotemporally. Sections 4.1 and 4.2 elaborate on this point.

While our work is not the first to emphasize the evaluation of FMs, it is unique on multiple fronts. Unlike Elevater (Li et al., 2022a)'s target of evaluating language-augmented FMs, we consider all FMs adaptable to video understanding. Unlike Perception Test (Authors, 2022)'s coverage of a broad spectrum of perception tasks, we focus on video, allowing us to cover various aspects of this vertical domain. Interestingly, many of our datasets also appear in InternVideo (Wang et al., 2022b), a video-oriented FM. However, we promote model adaptation methods as an inherent part of the evaluation protocol — a consistent set of diverse adaptation methods is necessary to provide FMs ample opportunities to expose their video understanding capabilities. Moreover, unlike InternVideo's focus on their single FM, we evaluate FMs developed by different research groups in an uniform experiment protocol — the first of its kind for visual and multimodal FMs, to the best of our knowledge.

Our main findings are as follows. First, task-specialized models still significantly outperform the six FMs studied in this work (see Figure 1), in sharp contrast to what FMs have achieved in natural language (OpenAI, 2022; Roberts et al., 2022) and image understanding (Radford et al., 2021; Yu et al., 2022; Chen et al., 2022). Hence, there is a need and tremendous opportunities to research video-focused FMs. Second, video-native FMs, whose pretraining data contains the video modality, are generally better than image-native FMs in classifying motion-rich videos, localizing actions in time, and understanding a video of more than one action. Third, the video-native FMs can perform well on video tasks under light adaptations to downstream tasks (e.g., freezing the FM backbones), while image-native FMs win in full end-to-end finetuning. This observation confirms that both tasks and adaptation methods matter when it comes to the evaluation of FMs.

## 2 RELATED WORK

**FMs.** One common type of FMs are Large Language Models (LLMs) trained to acquire generic, transferable, and diverse representations that can enable sample-efficient learning and knowledge transfer across a broad range of downstream tasks. FMs are often trained with simple self-supervised learning objectives such as predicting the next token in a sentence (e.g., GPT-3 (Brown et al., 2020), PaLM (Chowdhery et al., 2022)), or denoising the masked tokens (e.g., BERT (Devlin et al., 2018), UNILM (Dong et al., 2019), and BEiT (Bao et al., 2021)). An intriguing characteristic of FMs is their ability to gradually acquire new capabilities as the model grows and the training data size increases, despite being trained on simple learning objectives (Wei et al., 2022). For example, PaLM (Chowdhery et al., 2022; Anil et al., 2023), a massive LM with 540 billion parameters has started to show new capabilities in tasks such as explaining jokes, solving math, and performing common-sense reasoning when scaled to over 100B parameters.

In addition to self-supervised transformers, FMs in computer vision also encompass transformers specifically trained to align image-text paired data. These FMs use learning objectives include contrastive learning (e.g., CLIP (Radford et al., 2021)), denoising masked tokens (e.g., BEiT-3 (Wang et al., 2022a)), predicting the next token in a single modality (e.g., DALL-E (Ramesh et al., 2021)) or in the interleaved image-text sequence (e.g., Flamingo, KOSMOS-1 (Huang et al., 2023)). Recent FMs are also trained on a mixture of these objectives (e.g., CoCa (Yu et al., 2022), FLAVA (Singh et al., 2022), MAE (He et al., 2022)). For example, MAE combines autoencoder reconstruction objective jointly with the denoising objective (He et al., 2022) that was extended to video (Feichtenhofer et al., 2022; Tong et al., 2022). In our study, we choose six representative FMs (i.e., CoCa (Yu et al., 2022), CLIP (Radford et al., 2021), FLAVA (Singh et al., 2022), VideoMAE (Tong et al., 2022), VATT (Akbari et al., 2021), and InternVideo (Wang et al., 2022b)) due to their amendability on video understanding and accessibility of checkpoints.

**Evaluation of FMs.** As the mission of FMs is to enable sample-efficient knowledge transfer, the design of downstream tasks is critical to evaluate the capabilities and limitations of these models. The evaluation of FMs is pioneered by the NLP researchers. For example, GLUE (Wang et al., 2018a) and SuperGLUE (Wang et al., 2019) introduced a suite of tools for evaluating language understanding tasks. The authors utilized established public benchmarks and provided tools for evaluating, probing, and benchmarking pretrained FMs, allowing for a comparison to human baselines. ELEVATER (Li et al., 2022a) introduced this concept to vision FMs along with a toolkit for evaluating vision-language tasks, including knowledge augmentation, hyperparameter tuning, and three adaptation techniques. In parallel, there have been attempts to establish a diagnostic benchmark for perceptual understanding of the world. For instance, Perception Test (Authors, 2022) crowd-sourced 11K videos in which about 100 users performed scripted activities. This benchmark (Authors, 2022) comprises videos filmed by only about 100 participants, which may not provide the same level of domain coverage and diversity as the other FM evaluation works mentioned earlier.

**Evaluation of video FMs.** While some vision-language FMs have incorporated video tasks, their evaluation typically follows that of static images and neglects the unique aspects of video spatial-temporal modeling and reasoning. To our knowledge, no previous work has been solely dedicated to evaluating Video FMs. The closest work to ours are InternVideo (Wang et al., 2022b) and VideoMAE (Tong et al., 2022), which introduce new FMs and show their superiority over several dozen video datasets. There are two key differences to the prior works. First, our evaluation is video-centric using the tasks that require motion understanding or long-term temporal reasoning. Second, instead of promoting new video FMs, our work proposes no new models and is solely dedicated to evaluating current and future Video FMs in an impartial reproducible experimental setup. Concretely, our goal is to provide tools for probing and benchmarking FMs on motion tasks in various setting include using the parameter-efficient adapter.

## 3 TASKS AND ADAPTATION METHODS BOTH MATTER WHEN EVALUATING FMS

This section describes our video general understanding evaluation (VideoGLUE) benchmark. We first introduce the visual and multimodal FMs evaluated in this work. Then we discuss the video-focused downstream tasks and methods to adapt an FM to the tasks. The former concretizes the video

Table 1: Foundation models (FMs) studied in this work (MxM stands for Masked {Image, Language, or Video} Modeling).

| Foundation Model | Modality | Pretraining Data | Pretraining Objective |
|---|---|---|---|
| CoCa | Image + Text | JFT3B + ALIGN | Contrastive + Captioning |
| CLIP | Image + Text | WebImageText | Contrastive |
| FLAVA | Image + Text | PMD | Contrastive + MIM + MLM |
| VideoMAE | Video | K400 | MVM |
| InternVideo | Video | UnlabeledHybrid | MVM + Contrastive |
| VATT | Video + Audio + Text | HT100M | Contrastive |

Table 2: Summary of statistics, video properties, and data sources of each dataset. Tasks involved are spatiotemporal action localization (STAL), temporal action localization (TAL), and video classification (VC). Column "Num. videos" contains video examples in train/evaluation splits, respectively.

| Task | Dataset | Num. videos | Avg. length | Data source | Note |
|---|---|---|---|---|---|
| STAL | AVA v2.2 | $210,634\,/\,57,371$ | 15 mins | Movie | spatiotemporal, instance |
| | AVA-Kinetics | $354,201\,/\,91,919$ | 10 seconds | Web | spatiotemporal, instance |
| TAL | ActivityNet v1.3 | $10,002\,/\,4,926$ | 5-10 mins | Web | temporal |
| VC | Kinetics400 | $235,693\,/\,19,165$ | 10 seconds | Web | holistic, appearance |
| | Moments-in-Time | $791,246\,/\,33,898$ | 3 seconds | Web | holistic, appearance |
| | Sth-sth v2 | $168,913\,/\,24,777$ | 2-6 seconds | Crowd-source | holistic, motion |
| | Diving48 | $15,027\,/\,1,970$ | 5 seconds | Web | holistic, motion |
| | Charades | $7,811\,/\,1,814$ | 30 seconds | Crowd-source | multi-label, long-clip |

understanding capabilities we want to evaluate from an FM, while the latter provides various paths for an FM to showcase the corresponding capabilities.

### 3.1 FMs FOR VIDEO UNDERSTANDING

We are interested in examining which FMs are good at solving video tasks, what makes them better than others in the video domain, and how to best adapt them to video understanding. Table 1 shows the six FMs we gained access to via public repositories or personal communications.

### 3.2 VIDEO UNDERSTANDING TASKS

Like objects' role in image understanding, actions are the core of video understanding, leading us to select tasks and datasets that *recognize* and *localize* actions in time and space. Table 2 provides a quick summary. Next, we explain the rationale behind the particular choices of datasets and postpone the datasets' details to the supplementary materials.

#### 3.2.1 RECOGNIZING ACTIONS

**General actions.** We first include the action recognition datasets of Kinetics400 (K400) (Kay et al., 2017), Moments-in-Time (MiT) (Monfort et al., 2019), and Charades (Sigurdsson et al., 2016), considering their popularity that they are being complementary to each other. Regarding data sources, K400 videos are from Youtube, MiT draws videos from different Web venues, while Charades contains scripted videos. Regarding action labels, the datasets differ in granularities and real-life scenarios, a verb defines an action in MiT, K400 groups actions by verb-subject pairs, and Charades actions are about indoor activities. Regarding the average length, K400 and MiT videos are between 3 and 10 seconds, each with one action label, while Charades videos are about 30 seconds, each with multiple actions.

**Fine-grained motion-focused actions.** We also include Something-something-v2 (SSv2) (Goyal et al., 2017) and Diving48 (D48) (Li et al., 2018) as another two action recognition datasets, whose actions are fine-grained and motion-focused. SSv2 contains 174 human hand gestures as action labels, such as putting something into something, turning something upside down, and covering something

with something. D48 is all about competitive diving. Notably, the foreground objects' motion is a more significant discriminative cue than their appearance.

### 3.2.2 LOCALIZING ACTIONS

The videos in action recognition are trimmed, but actions could occur anywhere in a video in the wild. Hence, temporal and spatiotemporal action localization is also crucial to video understanding. Accordingly, we choose three datasets for the experiments: the action localization track of ActivityNet v1.3 (ANet) (Fabian Caba Heilbron & Niebles, 2015), Atomic Visual Actions (AVA) (Gu et al., 2018), and AVA-Kinetics (AVA-K) (Li et al., 2020). The last two require a model to localize (and recognize) actions in both time and space, and their underlying videos are movies and general YouTube videos, respectively.

### 3.3 ADAPTATION METHODS

In this section, we detail the task-specific neural architecture design and adaptation methods when applying FMs to downstream tasks.

#### 3.3.1 MODIFYING FM ARCHITECTURES FOR DOWNSTREAM TASKS

Given a $\text{FM}(\cdot)$, we can apply $\text{FM}(\cdot)$ to a video clip $C$ to extract a set of $k$ feature maps $\{F\}^k = \text{FM}(C), F \in \mathbb{R}^{n \times h \times w \times c}$, where $k$ is the number of endpoint layers from a FM, and $n, h, w, c$ are respectively a feature map's length, height, width, and number of channels.

For video classification tasks, we cast a feature map $F$ as $n \times h \times w$ tokens and aggregate them into a global representation using a learnable query token $\tau$ and lightweight cross-attention layers (Dosovitskiy et al., 2020). For spatiotemporal action localization, following the standard practice (Feichtenhofer et al., 2019; Tong et al., 2022), we first detect humans on key-frames using a human detector (Ren et al., 2015), producing a set of human bounding boxes $B$. We then apply the RoI pooling operation (Jaderberg et al., 2015) that takes both the feature map $F$ and box coordinates $B$ as inputs and outputs one feature vector per box as the query token, $\tau = \text{ROIPOOL}(F, B)$, followed by the same cross-attention layers as in video classification. For both groups of tasks, we stack a linear classifier on top of the task token's last-layer encoding for final classification:

$$p = \text{LINEARCLASSIFIER}(\text{CROSSATTENTION}(\tau, F)). \tag{1}$$

For temporal action localization, we first perform feature extraction in a sliding window manner, resulting in a sequence of globally average pooled features $\{\text{AVGPOOL}(F_1), \cdots, \text{AVGPOOL}(F_t)\}$ for each video. Following a popular choice of prior works (Alwassel et al., 2021; Ju et al., 2022; Liu et al., 2022), we employ G-TAD (Xu et al., 2020) as our task head for predicting the action category and its start and end timestamps.

#### 3.3.2 ADAPTING THE MODIFIED FMS' WEIGHTS FOR DOWNSTREAM TASKS

Adapting the modified FMs to a downstream task is to tune their weights. Then, we immediately have two basic adaptation strategies: 1) full finetuning to update all weights in the original FM plus the task head and 2) freezing FM weights and only updating newly added weights. The choice of the adaptation methods depends on specific application scenarios such as computation and memory constraints. We argue that an ideal FM should perform well across various adaptation methods to support the breadth of use cases.

**End-to-end finetuning.** End-to-end finetuning is the most common FM evaluation method for videos (Akbari et al., 2021; Feichtenhofer et al., 2022; Tong et al., 2022; Wang et al., 2022b), but it requires the deployment of a separate and possibly expensive FM for each downstream task. When finetuning all weights in the modified FMs, we limit cross-attention to a single transformer layer with 12 heads and hidden size 768. We vary learning rates and weight decays for each experiment to ensure every FM is configured to its best setup. Figure 2(a) illustrates this end-to-end finetuning.

**Frozen FM.** Linear probing and cross-attention based pooling over frozen FM features are routinely used to test the strength of the FM representation (Tong et al., 2022; Yu et al., 2022; Singh et al., 2022; He et al., 2022; Lin et al., 2022). In practice, adapting task-specific heads with a frozen FM

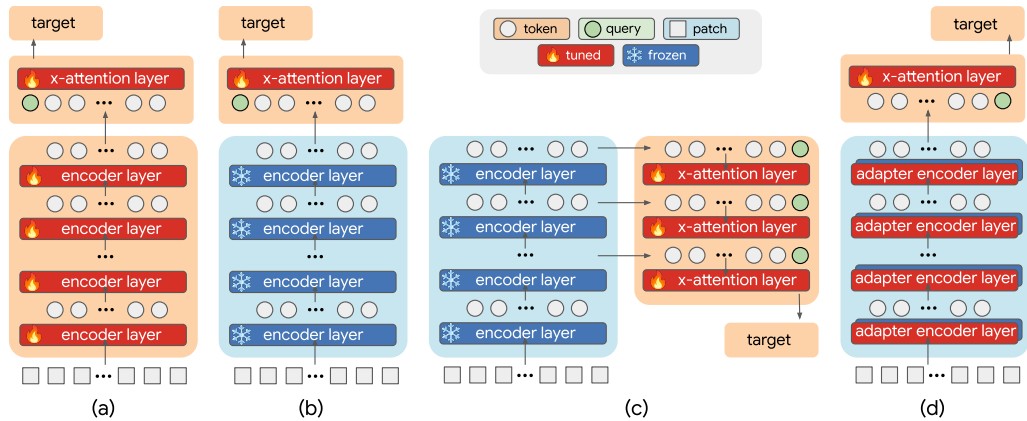

Figure 2: We study four adaptation methods to apply a foundation model (FM) to video understanding downstream tasks: (a) end-to-end finetuning, (b) frozen backbone evaluation, (c) frozen features with multi-layer attention pooler (MLAP), and (d) a low-rank adapter.

allows us to deploy the same FM for multiple tasks. If we use light-weight heads over the FM features, then a single FM inference can serve multiple tasks efficiently in terms of both compute and memory. To this end, we examine two variations with a frozen FM, one with a single cross-attention layer and the other with multiple layers. The first results in exactly the same model architectures as in end-to-end finetuning (Figure 2(b)), and the second allows us to leverage an FM's hierarchical features beyond its last endpoint layer (Figure 2(c)). First, the frozen features are extracted from the last $k$ layers, $F_{N-k+1}$, $F_{N-k+2}$, ..., $F_N$. Then, attention pooling is applied between a learnable token $\tau$ and the features $F_{N-k+1}$ using multi-head cross-attention (MHCA). The output of this layer serves as the query token for the next round of attention pooling with the features $F_{N-k+2}$. This process is repeated for $k$ rounds:

$$\tau_{N-k+1} = \text{MLP}(\text{MHCA}(\tau, F_{N-k+1}))$$
$$\tau_{N-k+2} = \text{MLP}(\text{MHCA}(\tau_{N-k+1}, F_{N-k+2}))$$
$$...$$
$$\tau_N = \text{MLP}(\text{MHCA}(\tau_{N-1}, F_N))$$

(2)

where $k = 4$ in our experiments, and the final classifier is $p = \text{LinearClassifier}(\tau_N)$.

**Frozen FM with a low-rank adapter**. Finally, we explore a frozen FM beyond the last $k$ layers using a low-rank adapter (Hu et al., 2021), which is a bottleneck architecture that projects a feature tensor into a low-dimensional space and then up-samples to the original space. The bottleneck space's dimension is 64 in our experiments. Inserting a few adapter layers with trainable weights $\{w\}$ into the pretrained FM while keeping all FM's weights frozen, the feature adapter is more parameter-efficient than end-to-end finetuning the whole network while achieving better performance than simply adding a task head to the frozen FM. Essentially, the adapter leads to a new $\widetilde{\text{FM}}$ with some trainable weights $\{w\}$: $\tilde{F} = \widetilde{\text{FM}}(C, \{w\})$, such that the output feature maps remain the same in shape as the original FM's output (Figure 2(d)). Hence, different pooling schemes and task heads aforementioned could be applied to the extracted feature map $\tilde{F}$. For simplicity, we still choose the single-layer cross-attention as the default task head due to its computation efficiency and performance.

The low-rank adaptation allows a single FM for multiple tasks, in contrast to the per-task models in end-to-end finetuning. However, it incurs a per-task forward pass at inference time, being less efficient than the task-specific heads over frozen features.

## 4 EXPERIMENTS

### 4.1 END-TO-END FINETUNING

Table 3 shows the end-to-end finetuning results of six FMs on eight datasets. We split the FMs into two groups based on their input modalities at the time of pretraining: CoCa, CLIP, and FLAVA are

Table 3: Evaluating FMs when adapted to video understanding tasks using end-to-end finetuning. We report the Top-1 accuracy on K400, MiT, D48 and SSv2, MAP on Charades and ANet, and mAP@IOU0.5 on AVA and AVA-K.

| | STAL | | TAL | VC (A) | | VC (M) | | VC (ML) | |
|---|---|---|---|---|---|---|---|---|---|
| Method | AVA | AVA-K | ANet | K400 | MiT | D48 | SSv2 | Charades | AVG |
| CoCa | **27.7** | **31.0** | – | **82.6** | **43.6** | **79.6** | 66.8 | 55.0 | 55.2 |
| CLIP | 27.1 | 28.9 | – | 81.0 | 39.0 | 75.7 | 46.6 | 54.3 | 52.8 |
| FLAVA | 22.0 | 25.6 | – | 79.1 | 38.3 | 72.0 | 61.1 | 48.6 | 49.4 |
| VideoMAE | 23.5 | 26.2 | – | 78.7 | 36.1 | 75.5 | 65.5 | 51.4 | 51.0 |
| InternVideo | 27.2 | 29.8 | – | 80.1 | 35.9 | 75.8 | **67.0** | 52.2 | 52.5 |
| VATT | 27.0 | 28.4 | – | 77.1 | 34.8 | 77.6 | 65.1 | **55.7** | 52.7 |
| Task-specialized | 42.3 RAFT | 38.9 RAFT | 37.5 PRN | 88.6 TubeViT | 42.7 UniformerV2 | 88.9 AIM | 68.7 MViT | 63.2 MoViNet | – |

Table 4: Evaluating FMs when adapted to video understanding using frozen features. Only weights in the task heads are updated using the downstream tasks' training sets.

| | STAL | | TAL | VC (A) | | VC (M) | | VC (ML) | |
|---|---|---|---|---|---|---|---|---|---|
| Method | AVA | AVA-K | ANet | K400 | MiT | D48 | SSv2 | Charades | AVG |
| CoCa | **23.3** | 24.7 | 33.0 | 73.1 | 32.0 | 34.1 | 41.5 | 8.8 | 31.2 |
| CLIP | 21.1 | **25.9** | 32.7 | **75.2** | **32.6** | 44.1 | 41.0 | 11.2 | 32.8 |
| FLAVA | 18.8 | 21.5 | 32.2 | 71.3 | 29.7 | 45.9 | 40.6 | 12.6 | 31.7 |
| VideoMAE | 16.0 | 19.9 | 33.0 | 65.1 | 23.0 | **59.5** | 53.9 | 11.3 | 32.6 |
| InternVideo | 13.4 | 15.7 | 33.3 | 69.3 | 26.3 | 55.6 | **58.2** | 13.0 | 33.1 |
| VATT | 20.3 | 22.2 | **35.3** | 75.1 | 32.1 | 49.7 | 57.8 | **33.3** | 39.1 |

image-native FMs, and VideoMAE, VATT, and InternVideo are video-native. The datasets span spatiotemporal action localization (STAL), video classification (VC), and temporal action localization (TAL). Note that we freeze FM weights in TAL because otherwise its full finetuning consumes excessive memory and computation. We draw the following observations from Table 3.

*FMs underperform task-specialized models on video tasks in general.* Table 3's last row collects the state-of-the-art results on the eight datasets, each obtained by a task-specialized model with comparable architecture or size to ours in the prior work. Specifically, those task-specialized models are RAFT (Rajasegaran et al., 2023), PRN (Wang et al., 2021), TubeViT (Piergiovanni et al., 2023), UniformerV2 (Li et al., 2022b), AIM (Yang et al., 2023), MViT (Fan et al., 2021) and MoViNet (Kondratyuk et al., 2021) respectively. All six FMs significantly underform the task-specialized models on the video tasks at the comparable model scale, indicating the lack of strong video-focused FMs. This observation is in sharp contrast to what FMs have achieved on natural language (OpenAI, 2022; Anil et al., 2023) and image understanding (Chen et al., 2022).

*Video-native FMs outperform image-native FMs on SSv2, Charades, and ANet* which require a model to reason along the time dimension: SSv2 actions are motion-rich, Charades has multiple actions per video, and ANet is about temporal action localization. These results strut the advantages of video-native FMs over image-native ones and, hopefully, prompt more efforts dedicating to the research of video-native FMs.

*CoCa performs the best among image-native FMs on the video tasks.* It actually gives rise to the highest accuracy on all datasets except SSv2, Charades, and ANet probably because CoCa, pretrained using image-text pairs, does not capture sufficient motion signals required for understanding SSv2, and it cannot handle Charades and ANet's complex, multiple actions per video.

## 4.2 FROZEN FMs

End-to-end finetuning is infeasible for some application scenarios due to FMs' rapidly growth in size and the consequent demands in computational resources. In the following, we evaluate frozen FMs with various adaptation methods. Tables 4, 5, and 6 are the results of adaptation with a single cross-attention layer, multiple cross-attention layers, and a low-rank adapter, respectively.

*CLIP generally performs the best among image-native frozen FMs (Tables 4 and 5), but CoCa catches up thanks to the low-rank adapter (Table 6).* It is worth noting that this ranking of image-native

Table 5: Evaluating FMs when adapted to video understanding using multi-layer attention pooler (MLAP), which takes multiple frozen features from an FM as inputs and map them hierarchically for the final task prediction. Only the multi-layer attention pooling layers are updated using the downstream tasks' training sets.

| | STAL | | TAL | VC (A) | | VC (M) | | VC (ML) | |
|---|---|---|---|---|---|---|---|---|---|
| Method | AVA | AVA-K | ANet | K400 | MiT | D48 | SSv2 | Charades | AVG |
| CoCa | 24.4 | 27.0 | 33.3 | 74.2 | 37.2 | 48.4 | 45.9 | 19.6 | 36.3 |
| CLIP | 27.7 | 29.6 | 33.9 | 77.1 | 39.0 | 55.8 | 50.1 | 41.5 | 43.3 |
| FLAVA | 21.3 | 23.2 | 32.4 | 71.5 | 34.5 | 58.5 | 43.1 | 38.2 | 39.3 |
| VideoMAE | 19.6 | 22.1 | 33.4 | 71.7 | 32.2 | 69.6 | 57.4 | 35.9 | 40.9 |
| InternVideo | 15.9 | 17.7 | 33.6 | 73.7 | 34.7 | 71.9 | 60.3 | 40.5 | 42.2 |
| VATT | 22.9 | 24.1 | 35.0 | 75.1 | 35.6 | 60.1 | 58.7 | 58.2 | 46.3 |

Table 6: The low-rank adapter results of FMs for video understanding. We only update the weights of the adapter and task head while keeping the original FMs' weights frozen.

| | STAL | | TAL | VC (A) | | VC (M) | | VC (ML) | |
|---|---|---|---|---|---|---|---|---|---|
| Method | AVA | AVA-K | ANet | K400 | MiT | D48 | SSv2 | Charades | AVG |
| CoCa | 26.6 | 28.7 | – | 80.9 | 41.4 | 67.1 | 56.1 | 45.8 | 49.0 |
| CLIP | 24.5 | 28.0 | – | 80.2 | 39.7 | 77.2 | 56.0 | 44.2 | 49.3 |
| FLAVA | 17.9 | 23.8 | – | 74.7 | 34.1 | 68.4 | 52.1 | 40.8 | 44.1 |
| VideoMAE | 16.6 | 23.3 | – | 73.6 | 30.6 | 76.0 | 61.4 | 43.0 | 45.9 |
| InternVideo | 19.2 | 25.5 | – | 75.5 | 31.3 | 73.6 | 63.9 | 46.2 | 47.7 |
| VATT | 22.3 | 25.8 | – | 75.0 | 36.5 | 68.9 | 63.5 | 53.5 | 49.9 |

frozen FMs differs from the ranking of image-native FMs in end-to-end finetuning. It seems that CLIP's endpoint features are more amendable to the video tasks than CoCa, but CoCa as a whole adapts better to video under both finetuning and the adapter. Hence, it is crucial to consider adaptation methods as an organic part of the evaluation of FMs to supply them various paths to demonstrate their capabilities.

*Video-native FMs are better than image-native FMs in understanding motion-rich SSv2 and D48, Charades that contain multiple actions per video, and ANet for temporal action localization.* This observation is about the same as the one under end-to-end finetuning. The image-native FMs is mainly superior on appearance-rich video datasets, where high-quality spatial perceptual features are the key. We conjecture that the vast image data empowering image-native FMs is more diverse in appearance than videos used to pretrain video-native FMs.

*Given frozen FMs, the low-rank adapter outperforms cross-attention layers, and multiple layers of cross-attention is better than a single cross-attention layer.* Many works (Caron et al., 2021; He et al., 2022) have shown features from different layers of a vision transformer have different attention maps. Hence, it is potentially beneficial to have an adaptation method to leverage multiple layers of a frozen FM. Table 5 reports the results with four cross-attention layers, whose average score per model (across different columns) is higher than that with a single cross-attention layer (Table 4) by 18% to 40%. The low-rank adapter (Table 6) further improves upon the cross-attention results partially because it explores all layers of a frozen FM.

*On average, image-native FMs outperform video-native FMs under end-to-end finetuning and the adapter, but it becomes the inverse in the other two adaptation methods.* The adapter experiment paired with end-to-end finetuning experiment reveal the fact that existing image-based FMs could be more easily adapted to video tasks when we could adjust the feature space of FMs, possibly caused by the large-scale higher quality image(-text) pretraining datasets. On the other hand, frozen feature experiments discussed above present us the inverse picture where video-based FM performs better. The seemingly paradox encourages more future research on bridging the gap on video-based pretraining with high-quality data and more effective modeling.

### 4.3 VIDEOGLUE SCORE: AN ATTEMPT TOWARDS RANKING FMS' VIDEO CAPABILITIES

In this section, we consolidate our studies of the FMs with different adaptation methods on a broad range of video tasks by considering their adaptation efficacies and efficiencies. Adaptation methods with different numbers of trainable weights lead to incompatible comparisons. Motivated

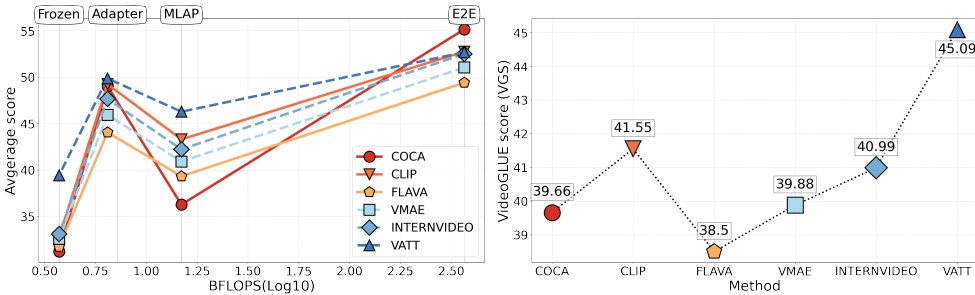

Figure 3: FMs are equipped with different adaptation methods. Left: For each adaptation method, we plot FMs' averaged scores across all video tasks vs. trainable FLOPs in a log scale. Right: We plot the overall VideoGLUE score (*VGS*) per FM.

by this, we propose a scalar measure, called VideoGLUE score (*VGS*), to capture an FM's overall adaptation performance on our video understanding tasks. While the VideoGLUE score may not be a perfect metric, it condenses multiple aspects of comparison into a scalar value, enabling a simplified comparison of FMs.

Taking the adaptation efficiency into account, we propose to use the trainable FLOPs to normalize an adapted FM's average score $s$ over all tasks. The trainable FLOPs are better than tunable weights because they allow our *VGS* to reflect both the model architecture's freedom and the input data's impact (e.g., sequence length) on downstream tasks. Formally, denoting by $\mathcal{S}_i$ an FM's average score over our video tasks under the $i$-th adaptation method and by $F_i$ the corresponding trainable FLOPs (in billion), we calculate the FM's *VGS* by

$$VGS = \sum_{i=1}^{N} w_i \mathcal{S}_i, \text{ where } w_i = \frac{\mathcal{A}_i}{\sum_{j=1}^{N} \mathcal{A}_j} \text{ and } \mathcal{A}_i = \frac{1}{\log_{10} F_i}, \tag{3}$$

where $N = 4$ is the number of adaptation methods, and $w_i \in [0, 1]$ weigh score $\mathcal{S}_i$ according to the trainable FLOPs $F_i$.

In Figure 3 we plot the averaged score achieved by each FM under each adaptation method, respectively, and compare their overall video understanding capabilities using the proposed *VGS*. The changes in FMs' ranking by different adaptation methods (see the left panel in Figure 3) reinforce that the adaptation methods matter and should be considered an organic part of the evaluation of FMs. On the right panel of Figure 3, we notice that the video-native FMs overall outperform image-native FMs on our video understanding tasks, achieving averaged *VGS* 41.98 vs. 39.90 respectively. This is intuitive as video-native FMs probably have a smaller domain gap to our tasks and are more capable of temporal and motion reasoning, which are important cues for video understanding. Zooming in to the individual FMs, we find that VATT, a video-native FM, is at the first place with *VGS* 45.1, followed by the image-native CLIP with *VGS* 41.6. This suggests that in-domain pretraining yields overall the best adaptation capability to video tasks, and image-native FMs could also achieve competitive results on many but not all video understanding tasks.

## 5 CONCLUSION

In this report, we study three image-based and three video-based foundation models and their adaptation capability on general video understanding tasks. Experiments are conducted on three hallmark video tasks, eight diverse datasets with four distinct adaption methods. Our study shows existing image-based FMs performs well on some appearance-rich video datasets, while video-based FMs tend to achieve better on motional and temporal reasoning. Four studied adaption methods curve different landscape, revealing the critical role of considering adaption methods as an organic part of evaluating FMs. Finally, we propose one single metric *VGS* to represent the video task adaptation efficiency of FMs. We hope our research provides useful resources for evaluating and analyzing video foundation models, and address the current gap in foundation model evaluation within the video domain.

SUPPLEMENTARY MATERIALS

We first discuss the limitations, ethcial concerns and broader impact of this work (Section A). We detail the datasets (Section B), models (Section C), and training setups (Section D) in the supplementary materials to improve this work's reproducibility. Besides, Section E includes more experimental studies to strengthen the main text.

## A  LIMITATION, ETHICAL CONCERN, AND BROADER IMPACT

**Limitation.** VideoGLUE covers various unimodal video tasks and could be strengthened by adding multimodal tasks like video question answering. We chose three representative FM adaptation methods and used them to provide as uniform experiment protocols for different FMs as possible. However, some of our observations could be flipped with the evolution of adaptation methods, which are an active research area. We proposed a scalar score, VideoGLUE score (VGS), to capture the efficacy and efficiency of an FM on video understanding. However, VGS might be dominated by one or a few datasets — when it becomes a serious issue, we should probably improve the score and/or retire the other datasets from future versions of VideoGLUE. Indeed, VGS is not a perfect score that covers all aspects of FMs in a comprehensive manner. For example, it does not account for an FM's memory usage, model size, model architecture, etc. We hope future research will lead to new metrics to complement VGS and a more comprehensive evaluation of FMs for visual tasks.

**Ethical concern.** We evaluate FMs on three video tasks, eight datasets in total. We select the tasks and datasets based on their popularity and representativeness. Although carefully designed, our benchmark inevitably inherited some ethical concerns from those datasets. For instance, many of the datasets are curated by crawling videos from the Internet, which do not proportionately represent the experiences of the global population and can potentially lead to biased evaluations of FMs. Moreover, the video datasets involve human daily activities, leading to privacy concerns about the human actors in the videos. How to evaluate FMs for video understanding in a fair and privacy-preserving manner could be an important direction for future research.

**Broader impact.** Our research reveals the need and tremendous opportunities to research video-first FMs by improving pretraining video data and methodologies. Our studies on different adaptation methods on versatile tasks confirms that both tasks and adaptation methods matter when it comes to the evaluation of FMs, shedding light on the already vibrant area of FM adaptations. Finally, we hope our research could inspire research on foundation models development and video understanding in general, along with their applications in the real world.

## B  VIDEO UNDERSTANDING DATASETS

### B.1  APPEARANCE-FOCUSED ACTION RECOGNITION

Video classification is a task of classifying videos into pre-defined labels, with the major focus on human actions.

Kinetics400 (Kay et al., 2017) (K400) is a large-scale, high-quality video dataset widely used as a standard video classification benchmark. It contains more than 250k video clips with annotations of 400 human daily actions. The actions are human focused and cover a broad range of classes including human-human interactions and human-object interactions. Although the video clips span 10 seconds on average, many studies (Sevilla-Lara et al., 2021; Wang et al., 2018b) have pointed out the task could be easily solved on the Kinetics datasets by inferring from the static objects appeared or background environment — motion information is less important than the visual appearance. Hence, we categorize Kinetics400 as an appearance-focused action classification dataset.

Moments-in-Time (Monfort et al., 2019) (MiT) is a large-scale video event classification dataset, with one million human annotated short video clips (around 3 seconds each). The temporal span corresponds to the averaged duration of human working memory and is a temporal envelope holding meaningful actions between people, objects, and phenomena. Videos in MiT are annotated with 339 most used verbs in the English vocabulary.

## B.2 MOTION-FOCUSED ACTION RECOGNITION

Videos contain much more commonsense knowledge than still images do, such as an object's motion patterns and the causal consequences of an action, just to name a few. However, appearance-based benchmarks do not evaluate a model's understanding of such commonsense knowledge, complex scenes, and situations. In observance of this, some video datasets have been proposed and studied in recent years with the focus on motions and common-sensing reasoning that are prosperous in video data.

Something-something v2 (Goyal et al., 2017) (SSv2) is a collection of around 200k videos of human performing pre-defined, basic actions with everyday objects. There are 174 unique labels in total depicting atomic hand manipulations, like putting something into something, turning something upside down or covering something with something. This dataset benchmarks a model's fine-grained understanding capability of object motions and scene changes by making the label space atomic-action-focused and background-invariant.

Diving48 (Li et al., 2018) (D48) is introduced to evaluate a model's dynamic reasoning capability. The video clips in this dataset are obtained by segmenting online videos of major diving competitions. In total, there are around 18k videos annotated with 48 classes. Because of its standardization, the diving scenario is purposefully chosen to avoid the scene, object, and person biases.

## B.3 MULTI-LABEL DAILY ACTION CLASSIFICATION

Most of current action classification datasets involve video clips with a clean snapshot of a single action. In contrast, humans perform daily complex activities step-by-step, simultaneously, or in an interleaving manner. Towards more comprehensive human daily activity reasoning, Charades (Sigurdsson et al., 2016) is introduced. Different from web-collected datasets whose contents are more structured, Charades is collected by crowd-sourcing from hundreds of actors recording their videos in their own homes, acting out casual everyday activities. Charades brings in more diversity into the video classification task due to its close-to-daily-life setting. Its videos are 30 seconds long on average and have multi-label annotations testing models' understanding of complex daily activities with multiple steps. Charades provides 110k videos with 157 action classes for training and evaluation.

## B.4 TEMPORAL ACTION LOCALIZATION

Natural long videos contain scene changes and semantic shifts, while most of the existing video benchmarks formulate problems to focus on trimmed video clips. Such a gap introduces evaluation bias as clip-level benchmarks could not reflect a model's temporal feature discriminativeness, which is of key importance to solve long-form video understanding tasks. To comprehend the study on foundation models' video capabilities, we include the temporal action localization (TAL) task in our evaluation. The task of TAL is to predict not only the action labels but also each action instance's temporal boundary in untrimmed videos. We adopt ActivityNet v1.3 (Fabian Caba Heilbron & Niebles, 2015) as the dataset for the TAL task, which contains $10,002$ untrimmed videos in training and $4,985$ in validation. The video length in this dataset is between 5-10 minutes. In total, there are 200 types of activities annotated.

## B.5 SPATIOTEMPORAL ACTION LOCALIZATION

Spatiotemporal Action Localization (STAL) is a person-centric task that asks a system to localize actors and predict their atomic actions (Barker & Wright, 1955; Gu et al., 2018) in a transitory duration.

In AVA (Gu et al., 2018), 15 minutes long movie clips are densely annotated at 1Hz. In the key frames, every person is localized using a bounding box and labels corresponding to actions being performed by the actor. The label vocabulary consists of 80 different atomic visual actions. There are 430 different movies in total.

AVA-Kinetics (Li et al., 2020) follows the same labeling protocol as AVA, while its data source comes from the Kinetics700 (Kay et al., 2017) video pool. The dataset contains over 230k clips annotated with the 80 AVA action classes for each of the humans in key frames.

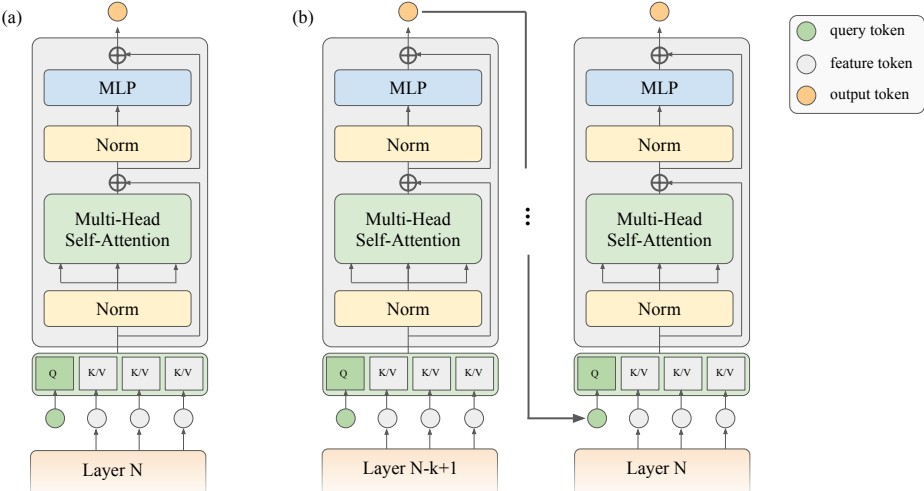

Figure 4: (a) Single-layer pooler head and (b) multi-layer attention pooling head for video classification and spatiotemporal action localization.

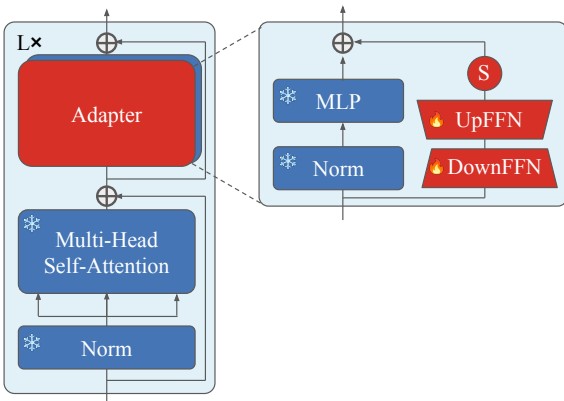

Figure 5: The adapter used in vision transformer. In the adapter layer, only the down-sample layer, up-sample layer, and the scaling factor are tunable. Between the down-sample layer and up-sample layer, an activation function is applied, which in our case is ReLU.

## C  MODEL DETAILS

### C.1  TASK HEAD ARCHITECTURES

In Figure 4, we plot the task heads used in our video classification and spatiotemporal action localization experiments, namely, the simple pooler head and multi-layer attention pooling head. For temporal localization, please refer to (Xu et al., 2020) for the task head's detailed architecture.

Figure 5 illustrates the encoder adapter layer's architecture. In the the adapter layer, only the down-sample layer, up-sample layer, and the scaling factor are tunable.

### C.2  IMAGE-TO-VIDEO ADAPTATION

Adapting image backbones to video tasks requires us to fuse the image embeddings at some point in the network and also introduce additional temporal information.

We consider two choices, early-fusion and late-fusion, and ablate them in the frozen feature setting in Table 7. In both early-fusion and late-fusion, we first apply the projection layer on each frame independently to embed pixel patches into embedding tokens. We then average-pool the embedding

Table 7: Early vs. late fusion on image-native FMs. In this experiment, the frozen feature with a single-layer pooler head is used.

| | K400 | | SSv2 | |
|---|---|---|---|---|
| Method | Early | Late | Early | Late |
| CoCa | 72.7 | 61.4 | 41.5 | 33.3 |
| CLIP | 70.5 | 75.2 | 38.1 | 41.0 |
| FLAVA | 67.9 | 71.3 | 40.4 | 40.6 |

Table 8: Ablation study on the temporal positional embedding for image-to-video adaption. We choose FLAVA (Singh et al., 2022) with the frozen feature setting in this experiment.

| Temporal Positional | VC (A) | | VC (M) | | VC (ML) |
|---|---|---|---|---|---|
| Embedding | K400 | MiT | D48 | SSv2 | Charades |
| ✗ | 71.3 | 29.7 | 41.6 | 30.3 | 10.7 |
| ✓ | 71.3 | 29.7 | 45.9 | 40.6 | 12.6 |

tokens from nearby frames to reduce the sequence length to $n \times h \times w$. In the early-fusion setting, we pass all tokens *together* to the image backbone to extract video features. In late-fusion, we pass each set of $h \times w$ tokens *independently* to the image backbone. Empirically, we find that the FLAVA (Singh et al., 2022) and CLIP (Radford et al., 2021) models do better with late-fusion while CoCa (Yu et al., 2022) does better with early-fusion.

Furthermore, we ablate the importance of temporal information using the frozen-features from FLAVA (Singh et al., 2022). In Table 8, we find that adding temporal positional embedding to the input is essential for D48 (Li et al., 2018), SSv2 (Goyal et al., 2017), and Charades (Sigurdsson et al., 2016) while not necessary for K400 (Kay et al., 2017) and MiT (Monfort et al., 2019). This supports our grouping that K400 and MiT are appearance-focused datasets.

Based on these findings, we use late-fusion for FLAVA (Singh et al., 2022) and CLIP (Radford et al., 2021) and early-fusion for CoCa (Yu et al., 2022). We add learnable temporal positional embeddings for all the image-native FMs.

## D    TASK-SPECIFIC HYPERPARAMETERS

In the following, we provide experiment settings and hyperparamters we used in this study. In Table 9, we list the hyperparameters we applied in the video classification task. In Table 10, we present the hyperparameters we used on spatiotemporal action localization. In Table 11, we present the hyperparameters we used on temporal action localization task.

We performed a greedy search on the learning rate and weight decay in all our experiments while keeping most other hyperparameters (e.g., data augmentation magnitude, dropout rate, drop path rate, etc.) consistent across different models and datasets. Specifically, we start with learning rate 1e-4 and weight decay 1e-5 and uniformly sample learning rates and weight decay factors with a rate of 5 and 10, respectively, centered around the starting points. After the first round, we pick the best-identified learning rate and weight decay factor as the new starting point and conduct another round of sampling with a rate of 2. We repeat another two to three rounds of hyperparameter search (with a rate of 2) until the model's performance converges. This process is a trade-off between computation costs and thoroughly examining an FM's performance under each experiment setup. The search ranges for the learning rate and weight decay are [4e-5, 2.5e-3] and [1e-6, 1e-4], respectively. We found that the learning rate is the most crucial factor when adapting an FM to downstream video understanding tasks.

Table 9: Experimental configurations for video classification tasks. We let learning rate and weight decay to be tunable per model to allow some flexibility for task adaptations.

| Config | Kinetics400 | Sth-sth v2 | MiT | Diving48 | Charades |
|---|---|---|---|---|---|
| batch size | 256 | 256 | 256 | 256 | 256 |
| training epochs | 150 | 50 | 50 | 100 | 50 |
| ViT sequence length | $8 \times 14 \times 14$ | $8 \times 14 \times 14$ | $8 \times 14 \times 14$ | $8 \times 14 \times 14$ | $8 \times 14 \times 14$ |
| **optimization** | | | | | |
| optimizer | AdamW | AdamW | AdamW | AdamW | AdamW |
| optimizer momentum | 0.9 | 0.9 | 0.9 | 0.9 | 0.9 |
| learning rate schedule | cosine decay | cosine decay | cosine decay | cosine decay | cosine decay |
| warmup ratio | 5% | 5% | 5% | 5% | 5% |
| **data augmentations** | | | | | |
| random horizontal flip | true | false | true | true | false |
| aspect ratio | (0.5, 2.0) | (0.5, 2.0) | (0.5, 2.0) | (0.5, 2.0) | (0.5, 2.0) |
| area ratio | (0.3, 1.0) | (0.3, 1.0) | (0.3, 1.0) | (0.3, 1.0) | (0.3, 1.0) |
| RandAug | (9, 0.5) | (9, 0.5) | - | - | - |
| MixUp | 0.8 | 0.8 | - | - | - |
| CutMix | 1.0 | 1.0 | - | - | - |
| **evaluation** | | | | | |
| multi-clips | 4 | 1 | 4 | 4 | 4 |
| multi-views | 3 | 3 | 3 | 3 | 3 |
| segment-based sample | false | true | false | false | false |

Table 10: Experimental configurations for spatiotemporal action localization.

| Config | AVA v2.2 | AVA-Kinetics |
|---|---|---|
| batch size | 256 | 256 |
| training epochs | 50 | 50 |
| ViT sequence length | $8 \times 16 \times 16$ | $8 \times 16 \times 16$ |
| **optimization** | | |
| optimizer | AdamW | AdamW |
| optimizer momentum | 0.9 | 0.9 |
| layer decay | 0.75 | 0.75 |
| learning rate schedule | cosine decay | cosine decay |
| warmup ratio | 5% | 5% |
| **data augmentations** | | |
| random horizontal flip | true | true |
| random scale | (0.5, 2.0) | (0.5, 2.0) |
| random color augmentation | true | true |

Table 11: Experimental configurations for temporal action localization.

| Config | ActivityNet v1.3 |
|---|---|
| batch size | 32 |
| training epochs | 10 |
| **feature extraction** | |
| fps | 15 |
| per-clip length | 16 |
| clip stride | 16 |
| **optimization** | |
| optimizer | AdamW |
| optimizer momentum | 0.9 |
| learning rate schedule | cosine decay |

# E    MORE STUDIES

## E.1    LARGE MODEL ADAPTATIONS

For the completeness of this report and reader's reference, in Table 12 we report experimental results under our settings with large FMs under two adaptation scenarios, namely, the frozen backbone with

Table 12: Evaluating large-scale FMs when using (a) frozen feature with a one-layer pooler head, and (b) low-rank adapter with frozen features. We report the Top-1 accuracy on K400, MiT, D48, SSv2 and MAP on Charades.

| Model | Method | VC (A) | | VC (M) | | VC (ML) |
|---|---|---|---|---|---|---|
| | | K400 | MiT | D48 | SSv2 | Charades |
| InternVideo-L | frozen | 78.6 | 33.7 | 69.6 | 67.4 | 20.9 |
| InternVideo-L | adapter | 81.5 | 40.3 | 85.8 | 70.9 | 54.2 |
| VideoMAE-v2-B/DL | frozen | 86.7 | 38.9 | 61.4 | 57.7 | 33.2 |
| VideoMAE-v2-B/DL | adapter | 86.0 | 41.8 | 82.3 | 66.6 | 53.8 |
| VideoMAE-v2-g | frozen | 59.7 | 20.7 | 42.5 | 44.2 | 12.7 |
| VideoMAE-v2-g | adapter | 80.8 | 35.9 | 85.3 | 68.2 | 55.5 |
| VideoMAE-v2-g/FT | frozen | 82.1 | 35.0 | 60.5 | 56.1 | 22.4 |
| VideoMAE-v2-g/FT | adapter | 85.2 | 42.5 | 84.6 | 70.6 | 58.6 |

Table 13: Benchmark FMs adaptation on video understanding tasks under sample-efficient transfer learning. This table shows Top-1 classification accuracy and the relative accuracy (shown in the bracket). Results are achieved by using frozen features with pooler head.

| Method | K400 | | | SSv2 | | |
|---|---|---|---|---|---|---|
| | 1% | 10% | 100% | 1% | 10% | 100% |
| CoCa | 27.1(37.8%) | 48.9(67.0%) | 73.1 | 5.6(13.4%) | 20.9(50.4%) | 41.5 |
| CLIP | 36.9(46.2%) | 66.8(83.6%) | 79.0 | 8.7(19.3%) | 25.1(55.5%) | 45.3 |
| FLAVA | 14.4(20.2%) | 35.8(50.3%) | 71.3 | 7.2(17.7%) | 14.3(35.3%) | 40.6 |
| VideoMAE | 15.5(23.9%) | 32.0(49.2%) | 65.0 | 13.7(25.4%) | 30.3(56.2%) | 53.9 |
| InternVideo | 20.4(29.5%) | 50.2(72.4%) | 69.3 | 19.5(33.6%) | 41.1(70.7%) | 58.2 |
| VATT | 34.1(45.4%) | 63.7(84.8%) | 75.1 | 12.9(22.4%) | 37.6(65.0%) | 57.8 |

pooler head and the low-rank adapter. VideoMAE-v2-B/DL (Wang et al., 2023) denotes the ViT-B model distilled from ViT-g on the Kinetics710 datasets[1]. VideoMAE-v2-g (Wang et al., 2023) is the model that pretrained on UnlabeledHybrid dataset, while VideoMAE-v2-g/FT (Wang et al., 2023) conducts further finetuning using supervised training on Kinetics710.

### E.2 SAMPLE-EFFICIENT TRANSFER LEARNING

A strong FM should be able to adapt to downstream tasks with a few training samples. In this section, we test the adaption ability of FMs in a sample-efficient transfer learning setting. Particularly, we freeze backbones and train a pooler head to adapt the FMs on K400 and SSv2. For either dataset, we sample 1% and 10% data from the training set uniformly for training and evaluate on the full evaluation dataset.

We show our experimental results in Table 13. To better understand the data efficiency, we also show the relative Top-1 accuracy for each model (shown in the bracket), which is defined as the ratio between accuracy with fewer training examples and the accuracy achieved using all the training data. A higher relative Top-1 accuracy means the performance of the model is closer to its "full" capacity under the sample-efficient setting. We notice that the best performed model on each dataset in fully fine-tuned model also performs best in the few-shot setting. Especially, CLIP (Radford et al., 2021) achieves 46.2% and 83.6% relative Top-1 accuracy on K400 using only 1% and 10% of the training data, respectively. On SSv2, InternVideo (Wang et al., 2022b) achieves 33.6% and 70.6% relative Top-1 accuracy with only 1% and 10% of the training data.

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
