# OpenReview forum: "VideoGLUE: Video General Understanding Evaluation of Foundation Models"
_ICLR.cc/2024/Conference — Submitted to ICLR 2024_

### Official Review · Reviewer_u4pP · 2023-10-30

**Soundness:** 2 fair
**Presentation:** 2 fair
**Contribution:** 2 fair
**Rating:** 5
**Confidence:** 4

**Summary:**

This paper introduces VideoGLUE, a comprehensive benchmark designed to evaluate video understanding capabilities of foundation models (FMs). VideoGLUE consists of three tasks (video classification, temporal localization, and spatio-temporal localization), eight datasets (K400, SSv2, D48, MiT, Charades, ANet, AVA, and AVA-K), and four adaptation methods (full finetuning, frozen backbone Eval., MLAP, and PETL w/ low-rank adapters). The authors also propose a scalar metric, VideoGLUE Score (VGS), that reflects the efficacy and efficiency of FMs. Comparative evaluations between image-native and video-native FMs on VideoGLUE reveal that 1) the existing FMs still lag behind task-specific models; 2) video-native FMs demonstrate superior temporal modeling capability than image-native counterparts; 3) video-native FMs outperform image-native FMs under light-weight adaptations, but not under full finetuning.

**Strengths:**

1.	The paper is well written and clearly motivated.
2.	The paper evaluates both image-native and video-native FMs on various video datasets and the unified evaluation protocols.
3.	The paper introduces various adaptation methods for both image and video FMs and the subsequent analysis underscores the critical role of adaptation in evaluation.

**Weaknesses:**

1.	VideoGLUE Score: (As the authors mentioned in Limitation Section) The proposed VideoGLUE score focuses solely on the computation-performance tradeoff, overlooking other valuable aspects, e.g., memory usage, model size, sample efficiency, model architecture difference. Given the significant influence of datasets and evaluation protocols on research directions, there is a concern that the current benchmark and scoring system might lead practitioners to overlook these crucial aspects of comprehensive video AI development..
2.	Selection of tasks: The benchmark is limted to three tasks, which may not fully capture the generalizability of FMs in video understanding. The inclusion of additional tasks, e.g., repetition counting, (long-term) action anticipation, human-object interaction, and so on,  could provide a more comprehensive evaluation.
3.	Selection of motion-focused benchmarks: Athough SSv2 is a widely acknowledged motion-focused dataset, existing literature [a, b] indicates that strong performance is achievable without motion modeling on SSv2 due to strong correlation between objects and action classes in SSv2. Utilizing Something-Else [a]as an alternative could offer a purer evaluation of motion modeling capability by breaking those spurious correlation. Furthermore, FineGym [c] provides hierarchical action labels allowing for a more granular assessment of a model modeling capability.

[a] Materzynska et al., Something-Else: Compositional Action Recognition with Spatial-Temporal Interaction Networks, CVPR, 2020. \
[b] Sun et al., Masked Motion Encoding for Self-Supervised Video Representation Learning, CVPR, 2023. \
[c] Shao et al., FineGym: A Hierarchical Video Dataset for Finegrained Action Understanding, CVPR, 2020.

**Questions:**

1. Is there any reason why image-native FM CoCa outperforms other FMs on D48, which mitigates static contextual biases but focuses on motion?

---

> ### Author Response · Authors · 2023-11-16
> **Thank you for your review**
>
> We hope our response addresses your concerns. Should you have additional questions, we are readily available for further discussion.
>
> * Re W1 "The proposed VideoGLUE score focuses solely on the computation-performance tradeoff."
>
> As reviewer 1CiA commented, “The suggested VideoGLUE score can be an interesting **initial** way to start a comparison and discussion of the strength of foundation models”. We acknowledge that it is only an initial attempt to score FMs by their video understanding capabilities, and it is not comprehensive as we discussed the limitations (more details in Appendix A).
>
> Nonetheless, it captures the essence and can help the design of new metrics in future work. In particular, we design VideoGLUE score (VGS) as the model performance weighted by the number of tunable parameters, which are arguably the two most important factors when evaluating an FM [1, 2].
>
> [1] Bommasani, R., Hudson, D.A., Adeli, E., Altman, R., Arora, S., von Arx, S., Bernstein, M.S., Bohg, J., Bosselut, A., Brunskill, E. and Brynjolfsson, E., 2021. On the opportunities and risks of foundation models. arXiv preprint arXiv:2108.07258.
>
> [2] Li, C., Liu, H., Li, L., Zhang, P., Aneja, J., Yang, J., Jin, P., Hu, H., Liu, Z., Lee, Y.J. and Gao, J., 2022. Elevater: A benchmark and toolkit for evaluating language-augmented visual models. Advances in Neural Information Processing Systems, 35, pp.9287-9301.
>
> * Re W2 "Selection of tasks are limited to three tasks."
>
>  We appreciate the reviewer’s suggestion, and we have to leave them to future work from the community. As you may notice, completing the evaluation on one set of metrics in our paper requires more than one hundred experiments (six foundation models, four adaptation methods over eight datasets), not to mention the hyperparameter tuning and other explorations. Hence, we indeed made hard decisions about which to include in VideoGLUE.
>
> We are working on a new video task/dataset. Should bandwidths permit, we will work on the second version of VideoGLUE by including the new task and investigating the tasks the reviewer suggested.
>
> * Re W3 "Selection of motion-focused benchmarks."
>
> We did consider Something-Else against SSv2 and FineGym against Diving and decided to go with SSv2 and Diving because they are well received by the community. We can swap them in the future if their drawbacks become an obstacle for fair evaluations.
>
> * Re Q1 "Is there any reason why image-native FM CoCa outperforms other FMs on D48"
>
> We note that although image-native CoCa and CLIP perform well on D48 in the end-to-end fine-tuning setting and the adapter setting, respectively, video-native FMs perform better in the frozen feature setting and the multi-layer attention pooler setting. Because D48 is small, it is unlikely to provide sufficient supervision for end-to-end fine-tuning the spatiotemporal weights in the video-native FMs. However, it still signifies the video-native FMs’ motion understanding capabilities when the backbones are frozen.

---

> > ### Author Response · Authors · 2023-11-22
> > **Look forward to your feedback**
> >
> > Dear Reviewer,
> >
> > As the reviewer-AC discussion phase will end in a day, we cordially seek feedback on our response to your reviews. Please let us know if there are outstanding issues, and we are eager to address them promptly. Your constructive input is highly valued, and we appreciate your commitment to improving our manuscript.
> >
> > Thank you for your time and consideration, Authors

---

> ### Comment · Reviewer_u4pP · 2023-11-23
> **Response to the authors**
>
> > As reviewer 1CiA commented, “The suggested VideoGLUE score can be an interesting initial way to start a comparison and discussion of the strength of foundation models”. We acknowledge that it is only an initial attempt to score FMs by their video understanding capabilities, and it is not comprehensive as we discussed the limitations (more details in Appendix A).
> Nonetheless, it captures the essence and can help the design of new metrics in future work. In particular, we design VideoGLUE score (VGS) as the model performance weighted by the number of tunable parameters, which are arguably the two most important factors when evaluating an FM [1, 2].
>
> I appreciate the efforts put into developing the VideoGLUE score as an initial step in evaluating video foundation models (FMs) as the Reviewer1CiA mentioned. However, IMHO, VideoGLUE requires further refinement to establish a comprehensive and clear benchmark, **especially considering its role not just as a method but as a benchmark and evaluation protocol**.
> For example, GLUE(NeurIPS’19) and SuperGLUE(ICLR’19), which is an improved version of GLUE, were published in the same year. The significant difference in their citations underscores the importance of a benchmark’s initial comprehensiveness and reception. As the first of its kind, VideoGLUE must ensure a robust and comprehensive foundation to garner significant attention and use.
>
> > We appreciate the reviewer’s suggestion, and we have to leave them to future work from the community. As you may notice, completing the evaluation on one set of metrics in our paper requires more than one hundred experiments (six foundation models, four adaptation methods over eight datasets), not to mention the hyperparameter tuning and other explorations. Hence, we indeed made hard decisions about which to include in VideoGLUE.
>
> Concerning the extensive requirements of video experiments, I understand the constraints of computational resources. However, this limitation raises concerns about the benchmark’s ability to assure the generalizability of video FMs. If not, the authors at least should’ve justified how the three chosen tasks can guarantee the measure of generalizability of video FMs and why other video tasks, e.g., repetition counting, or fine-grained motion understanding, are excluded.
>
> > We did consider Something-Else against SSv2 and FineGym against Diving and decided to go with SSv2 and Diving because they are well received by the community. We can swap them in the future if their drawbacks become an obstacle for fair evaluations.
>
> I respectfully disagree with the author's statement, especially, “We can swap them in the future if their drawbacks become an obstacle for fair evaluations.” at two points. First, the problem of spurious correlation of SSv2 has already been raised a couple of years ago, and thus, VideoGLUE should reflect current understanding and criticisms. Second, changing datasets in benchmarks is not trivial, as practitioners developing video models might rely on existing VideoGLUE scores for benchmarking, making any alterations challenging for fair comparisons.
>
> In summary, while acknowledging VideoGLUE’s initial contribution, I urge consideration of its comprehensiveness, justification of dataset selection, and the implications of changing benchmarks in the future for a fair and effective evaluation of video FMs. I thus keep my rating.

---

### Official Review · Reviewer_oktk · 2023-10-31

**Soundness:** 3 good
**Presentation:** 3 good
**Contribution:** 2 fair
**Rating:** 5
**Confidence:** 4

**Summary:**

This paper assesses the video understanding capabilities of current foundation models using a novel experimental protocol that spans multiple tasks, datasets, and adaptation methods. Additionally, it introduces a scalar VideoGLUE score to gauge both the efficacy and efficiency of a foundation model. Drawing from the experimental outcomes, the paper unveils several interesting findings.

**Strengths:**

1.	The motivation is both clear and justified. There is a pressing need in the community to establish a benchmark for assessing the video understanding capabilities of foundation models.
2.	The introduced VideoGLUE benchmark evaluates foundation models across various dimensions such as tasks, datasets, and adaptation methods.
3.	This paper highlights three interesting findings into the video understanding capabilities of current foundation models.
4.	The paper is well written and easy to follow.

**Weaknesses:**

1.	This paper analyzes six foundational models, varying in size, pre-training data, and objectives. The diversity of these settings compromises the comparability of the experimental results, rendering the conclusions less reliable.
2.	This paper introduces a scalar VideoGLUE score to assess FM's efficacy and efficiency by averaging performance scores across four adaptation methods. Yet, the rationale behind the metric's design appears arbitrary. It's unclear why this particular weighted score, derived from the specified datasets and adaptation methods, is indicative of FMs' video understanding capabilities.

**Questions:**

1.	In Table 3, the task-specialized model UniformerV2 registers a score of 42.7 on the MIT dataset, which trails the 43.6 scored by CoCa. This observation contradicts the assertion that "All six FMs underperform task-specialized models on video tasks."
2.	In Figure 3, CoCa achieves an average score of approximately 36 under the MLAP adaptation. However, Table 5 lists the average score for CoCa as 45.9. Why aren't they consistent?

---

> ### Author Response · Authors · 2023-11-16
> **Thank you for your comments!**
>
> We hope our response addresses your concerns. Should you have additional questions, we are readily available for further discussion.
>
> * Re W1 "The diversity of these settings compromises the comparability of the experimental results, rendering the conclusions less reliable."
>
> We acknowledge the reviewer's comment. In our experiment design, we try to compare FMs fairly, so that we strictly control our comparison in the main paper to the same model size, ViT-B, and resolutions and other hyperparameters like augmentation strategy. Pre-training data and objectives are not controllable and we note that it is also happening to the evaluation of large language models. While we cannot control the FMs' diverse pretraining settings, this work tries to standardize the evaluation so that we can examine "which FMs are good at solving video tasks, what makes them better than others in the video domain, and how to best adapt them to video understanding" (quoted from Section 3.1 in the paper).
>
> * Re W2 "design of VideoGLUE score":
>
> As reviewer 1CiA commented, “The suggested VideoGLUE score can be an interesting **initial** way to start a comparison and discussion of the strength of foundation models”. We acknowledge that it is only an initial attempt to score FMs by their video understanding capabilities, and it is not comprehensive as we discussed the limitations (more details in Appendix A). Nonetheless, it captures the essence and can help the design of new metrics in future work. In particular, we design VideoGLUE score (VGS) as the model performance weighted by the number of tunable parameters, which are arguably the two most important factors when evaluating an FM [1, 2].
>
> [1] Bommasani, R., Hudson, D.A., Adeli, E., Altman, R., Arora, S., von Arx, S., Bernstein, M.S., Bohg, J., Bosselut, A., Brunskill, E. and Brynjolfsson, E., 2021. On the opportunities and risks of foundation models. arXiv preprint arXiv:2108.07258.
>
> [2] Li, C., Liu, H., Li, L., Zhang, P., Aneja, J., Yang, J., Jin, P., Hu, H., Liu, Z., Lee, Y.J. and Gao, J., 2022. Elevater: A benchmark and toolkit for evaluating language-augmented visual models. Advances in Neural Information Processing Systems, 35, pp.9287-9301.
>
> * Re Q1 "score on the MIT dataset":
>
> Thanks for pointing this out. We will modify the statement to “Foundation models underperform task-specialized models on video tasks in general — MIT is the only exception.”
>
> * Re Q2 " Why Figure 3 and Table 5 CoCa scores are consistent?"
>
>  Thank you for catching this! We made a typo in Table 5. Figure 3 is the correct one. We have updated the manuscript accordingly.

---

> > ### Comment · Reviewer_oktk · 2023-11-21
> >
> > Thank you for your response. Despite your clarifications, I remain convinced that this submission is not yet suitable for publication as a conference paper at ICLR. As mentioned in my initial review, the necessity for a comprehensive and fair comparison of video foundation models is imperative for thorough evaluation. The present one, in its current form, appears more apt for inclusion in a technical report, which could be valuable for users in selecting appropriate models. Additionally, as pointed by Reviewer u4pP, the current VideoGLUE score lacks comprehensiveness, potentially leading to misconceptions in the development of video AI technology.

---

> > > ### Author Response · Authors · 2023-11-22
> > > **Response about the system-level comparison and score**
> > >
> > > Thank you for the quick response! We appreciate the opportunity to further discuss with you the two points you emphasized.
> > >
> > > > As mentioned in my initial review, the necessity for a comprehensive and fair comparison of video foundation models is imperative for thorough evaluation. (Initial review: This paper analyzes six foundational models, varying in size, pre-training data, and objectives. The diversity of these settings compromises the comparability of the experimental results, rendering the conclusions less reliable.)
> > > >
> > >
> > > Some of our team members had exactly the same concern, but we eventually decided to align with the practice of GLUE, assessing LLMs at the system level, due to the following considerations.
> > >
> > > Given our focus on video **general** understanding evaluation, a system-level assessment is pragmatically the most effective approach. First, many models are not open to us, so we have to instead make our evaluation standardized and publicly accessible. Second, we have no control over FMs’ pretraining settings, and the cost of re-training them in a fair setting is prohibitive. Third, given that GLUE accelerated the development of LLMs and remains helpful despite its system-level evaluation, we hold the same expectation for VideoGLUE.
> > >
> > > We acknowledge the merits of a fair comparison by accounting for the pretraining data and others, but our chosen methodology offers advantages. It grants model developers the freedom to conduct individualized data research, a fundamental, intrinsic aspect of any FM.
> > >
> > > We appreciate your recognition of the work’s potential value "for users in selecting appropriate models". As the paper tells in Section 3.1, we hope to provide some clues about "which FMs are good at solving video tasks, what makes them better than others in the video domain, and how to best adapt them to video understanding."
> > >
> > > Lastly, we emphasize that our work represents the most comprehensive evaluation of FMs in video understanding to date. In our selection, we cover FMs with image-based and video-based inputs (e.g., CLIP and VATT), pretrained with contrastive, generative, and masked modeling objectives (e.g., CLIP, CoCa and VideoMAE), plus the careful design and selection of downstream tasks and adaptation methods.
> > >
> > > > Additionally, as pointed by Reviewer u4pP, the current VideoGLUE score lacks comprehensiveness, potentially leading to misconceptions in the development of video AI technology.
> > > >
> > >
> > > As reviewer 1CiA commented, “The suggested VideoGLUE score can be an interesting **initial** way to start a comparison and discussion of the strength of foundation models”.
> > >
> > > We design VideoGLUE score (VGS) as the model performance weighted by the number of tunable parameters, which are arguably the two most important factors when evaluating an FM [1, 2]. The design of the score is extensible, so one should be able to easily incorporate the other factors (e.g., memory, model architecture, etc.) in future work.
> > >
> > > [1] Bommasani, R., Hudson, D.A., Adeli, E., Altman, R., Arora, S., von Arx, S., Bernstein, M.S., Bohg, J., Bosselut, A., Brunskill, E. and Brynjolfsson, E., 2021. On the opportunities and risks of foundation models. arXiv preprint arXiv:2108.07258.
> > >
> > > [2] Li, C., Liu, H., Li, L., Zhang, P., Aneja, J., Yang, J., Jin, P., Hu, H., Liu, Z., Lee, Y.J. and Gao, J., 2022. Elevater: A benchmark and toolkit for evaluating language-augmented visual models. Advances in Neural Information Processing Systems, 35, pp.9287-9301.
> > >
> > > **We look forward to any further feedback or suggestions you may have and appreciate your time and consideration.**

---

### Official Review · Reviewer_geS9 · 2023-10-31

**Soundness:** 3 good
**Presentation:** 3 good
**Contribution:** 3 good
**Rating:** 6
**Confidence:** 3

**Summary:**

This work aims to assess the video understanding capabilities of existing foundation models. The authors have employed a meticulous experimental protocol that encompasses three hallmark tasks: action recognition, temporal localization, and spatiotemporal localization. These tasks are evaluated across eight datasets that are well-regarded by the research community. The initial context suggests that the paper's focus is on the comprehensive evaluation of video understanding in foundational models using a variety of tasks and datasets.​

**Strengths:**

1. Evaluating Foundation Models (FMs): The paper emphasizes the complexity involved in evaluating FMs, particularly because they are designed as "generalists" that learn meta-knowledge across tasks. This highlights the need for a standardized evaluation procedure, which this paper aims to provide.

2. VideoGLUE Protocol: The proposed evaluation protocol provides a structured approach to evaluate FMs on video understanding, encompassing various tasks, datasets, and model adaptation methods. This could serve as a benchmark for future research.

**Weaknesses:**

1. Different datasets emphasize varied aspects in video tasks; for instance, SSV2 focuses on motion, while Kinetics is more context-centric. How does VideoGLUE address the differences among these diverse datasets?

2. While the study delves into transformer-based Foundation Models (FMs), is there any comprehensive analysis or comparison involving 3D-CNN or even 2D-CNN based FMs?

**Questions:**

See Weaknesses

---

> ### Author Response · Authors · 2023-11-16
> **Thank you for your comments**
>
> We thank the reviewer for the detailed comments and valuable suggestions. We hope our response addresses your concerns. Should you have additional questions, we are readily available for further discussion.

---

> ### Author Response · Authors · 2023-11-16
> **Response to reviewer geS9's comments**
>
> * Re "How does VideoGLUE address the differences among these diverse datasets?"
>
> We choose a diverse set of video datasets/tasks on purpose. These diverse video datasets/tasks help us probe into FMs’ video understanding capabilities from different perspectives. For example, SSv2 and Kinetics emphasize motion and appearance, respectively. Video classification and temporal action localization test an FM’s holistic understanding versus temporal reasoning.
>
> * Re "is there any comprehensive analysis or comparison involving 3D-CNN or even 2D-CNN based FMs?"
>
> We followed the following definition of foundation models: “any model that is trained on broad data that can be adapted (e.g., finetuned) to a wide range of downstream tasks” [1] to choose the most representative image- and video-based foundation models. Interestingly, the resultant FMs happen to be all transformer-based, potentially because the model developers chose the transformer architecture considering its better scalability than CNNs [2, 3].
>
> [1] Bommasani, R., Hudson, D.A., Adeli, E., Altman, R., Arora, S., von Arx, S., Bernstein, M.S., Bohg, J., Bosselut, A., Brunskill, E. and Brynjolfsson, E., 2021. On the opportunities and risks of foundation models. arXiv preprint arXiv:2108.07258.
>
> [2] Dosovitskiy, A., Beyer, L., Kolesnikov, A., Weissenborn, D., Zhai, X., Unterthiner, T., Dehghani, M., Minderer, M., Heigold, G., Gelly, S. and Uszkoreit, J., 2020. An image is worth 16x16 words: Transformers for image recognition at scale. arXiv preprint arXiv:2010.11929.
>
> [3] https://twitter.com/giffmana/status/1717995379034071483

---

> > ### Author Response · Authors · 2023-11-22
> > **Look forward to your feedback**
> >
> > Dear Reviewer,
> >
> > As the reviewer-AC discussion phase will end in a day, we cordially seek feedback on our response to your reviews. Please let us know if there are outstanding issues, and we are eager to address them promptly. Your constructive input is highly valued, and we appreciate your commitment to improving our manuscript.
> >
> > Thank you for your time and consideration, Authors

---

### Official Review · Reviewer_1CiA · 2023-10-31

**Soundness:** 3 good
**Presentation:** 4 excellent
**Contribution:** 2 fair
**Rating:** 6
**Confidence:** 5

**Summary:**

The paper studies video understanding capabilities of several existing foundation models. For this purpose, these models are evaluated on three primary video tasks: action recognition, temporal localization, and spatiotemporal localization. There are also three main findings from this study. First, task-specialized models still outperform foundation models; second, video-native foundation models are better than image-native models for complex and motion-rich videos; and third, light adaptations are enough for video-native models while full fine-tuning is better for image-native models. In addition, there is also an attempt to suggest one metric that resembles the strength of the foundation model.

**Strengths:**

1) The paper has an excellent presentation with clear discussion and motivation for the work done. The visualizations are very nice and they further facilitate the text of paper and experiments.
2) Evaluating six foundation models on three main video tasks and eight datasets is a valuable contribution to the research community. It is even more valuable because different adaptions are also considered (fine-tuning, low-rank adapter, and others). Experiments are performed thoroughly with good analysis and discussion. The obtained conclusions can be important to the next iteration of the development of foundation models.
3) The suggested VideoGLUE score can be an interesting initial way to start a comparison and discussion of the strength of foundation models. It seems that the scores for the studied models are quite close to each other and still far from the perfect one. So we can think that we are only starting scretching the cover of this area.

**Weaknesses:**

1) Despite the contribution of suggesting a VideoGLUE score, there are no novel models, modifications, or datasets presented in the paper. In my opinion, it would be very beneficial to develop at least one of them as an additional contribution to make the strongest possible paper.
2) The list of studied foundation models is not as comprehensive as potentially can be. I understand, that not models are publicly available but it would be very interesting and important to make even stronger conclusions by including some of video-based models developed on top of CLIP paradigm. Some of the examples are: "VideoCLIP: Contrastive Pre-training for Zero-shot Video-Text Understanding", Xu et al., EMNLP 2021; "Expanding Language-Image Pretrained Models for General Video Recognition", Bolin et al., ECCV 2022.
3) Also, there is a rapid development of adaptation techniques that could be also very nice to be included in the analysis. Some of the recent examples are: "AIM: Adapting Image Models for Efficient Video Action Recognition", Yang et al., ICLR 2023; "Frozen CLIP Models are Efficient Video Learners", Lin et al., ECCV 2022; "ST-Adapter: Parameter-Efficient Image-to-Video Transfer Learning", Pan et al., NeurIPS 2022. The listed papers are mostly adapters from images to videos, but it would be still great to have them in the analysis for some of the applicable foundation models.

**Questions:**

No other questions except those listed in the "Weaknesses" section.

---

> ### Author Response · Authors · 2023-11-16
> **Response to reviewer 1CiA's comments**
>
> We thank reviewer for the detailed comments and valuable suggestions. We hope our response addresses your concerns. Should you have additional questions, we are readily available for further discussion.
>
> * Re W1 "no novel models, modifications, or datasets presented in this paper":
>
> We appreciate the reviewer for this invaluable suggestion as we are actively working on a new video task/dataset. We cannot include it here because of the high workload involved. As one may notice, completing this work's evaluation on one set of metrics in our paper requires more than one hundred experiments (six foundation models, four adaptation methods over eight datasets), not to mention the hyperparameter tuning and other explorations. Including new models, modifications, or datasets would be great complements to this project but also require extensively more work. We will highlight this limitation in the revised paper.
>
> * Re W2 "The list of studied foundation models is not as comprehensive as potentially can be":
>
> Regrettably, we have lost the priority of computing resources for this project, and we are unable to provide further results for other models. Nevertheless, the paper has included two “video-based models developed on top of CLIP paradigm”: InternVideo and VATT. Especially, VATT is more or less a “superset” of VideoCLIP suggested in the review because VATT’s pretraining data and method cover VideoCLIP’s. Similarly, InternVideo can be viewed as a “superset” of the second paper mentioned in the review.
>
> * Re W3 "other adaptation techniques":
>
> Thanks for pointing these out. In our exploration phase, we tried multiple adapters, including the “ST-Adapter” suggested by the reviewer. However, we were unable to reproduce ST-Adapter’s results reported in the original paper. In contrast, LoRA was the most stable and widely embraced one with superior performance, so we chose to report LoRA in the paper.

---

> > ### Author Response · Authors · 2023-11-22
> > **Looking Forward to Your Feedback**
> >
> > Dear Reviewer,
> >
> > As the reviewer-AC discussion phase will end in a day, we cordially seek feedback on our response to your reviews. Please let us know if there are outstanding issues, and we are eager to address them promptly. Your constructive input is highly valued, and we appreciate your commitment to improving our manuscript.
> >
> > Thank you for your time and consideration, Authors

---

> > > ### Comment · Reviewer_1CiA · 2023-11-22
> > > **Thank you**
> > >
> > > Thank you for your response. I understand the limitations of the available resources and think the paper already contains a high value for the research community. However, my comments and comments from other reviewers point out that the paper can be even better and stronger. It seems that the authors also understand that, and they already know the right direction to improve the paper. I am not against accepting the paper in its current state however having more time to further improve the paper can be very beneficial for the authors and for the research community to have an even stronger comparison of foundation models.

---

### Author Response · Authors · 2023-11-16
**Thank you reviewers for your review**

We thank the reviewers for the detailed comments and valuable suggestions. We hope our response addresses your concerns. Should you have additional questions, we are readily available for further discussion.

---

### Author Response · Authors · 2023-11-21
**Looking Forward to Your (reviewers') Feedback**

Dear Reviewers,

As the reviewer-AC discussion phase will end in a couple of days, we cordially seek feedback on our response to your reviews. Please let us know if there are outstanding issues, and we are eager to address them promptly. Your constructive input is highly valued, and we appreciate your commitment to improving our manuscript.

Thank you for your time and consideration,
Authors

---

### Meta-Review · Area_Chair_DvZF · 2023-12-09

**Metareview:**

The meta-reviewer has carefully read the paper, reviews, rebuttals, and discussions between authors and reviewers. The meta-reviewer agrees with the reviewers that this paper is a bit below the bar of ICLR. The paper presents VideoGLUE, a benchmark created to assess the video understanding of foundation models (FMs). It encompasses three tasks (video classification, temporal localization, and spatiotemporal localization), eight datasets, and four adaptation methods (including full finetuning and low-rank adapters). A metric, VideoGLUE Score (VGS), is proposed to measure the performance and efficiency of FMs. However, the meta-reviewer agrees with the reviewers on the negative side, where the VideoGLUE Score is criticized for focusing only on computational performance, the benchmark includes tasks that may not fully reflect the versatility of FMs in video understanding. A positive reviewer 1CiA even points out that the paper can be further improved, "having more time to further improve the paper can be very beneficial for the authors and for the research community to have a stronger comparison of foundation models." Given the comments from all reviewers, the meta-review feels not confident to recommend an acceptance at the moment.

**Justification For Why Not Higher Score:**

N/A

**Justification For Why Not Lower Score:**

N/A

---

### Decision · Program_Chairs · 2024-01-16

Reject